# Ectopic Expression of *Gs5PTase8*, a Soybean Inositol Polyphosphate 5-Phosphatase, Enhances Salt Tolerance in Plants

**DOI:** 10.3390/ijms21031023

**Published:** 2020-02-04

**Authors:** Qi Jia, Song Sun, Defeng Kong, Junliang Song, Lumei Wu, Zhen Yan, Lin Zuo, Yingjie Yang, Kangjing Liang, Wenxiong Lin, Jinwen Huang

**Affiliations:** 1Key Laboratory for Genetics Breeding and Multiple Utilization of Crops, Ministry of Education/College of Agriculture, Fujian Agriculture and Forestry University, Fuzhou 350002, China; sunsong07@163.com (S.S.); kongyuanyuan00210@163.com (D.K.); xgrz@163.com (J.S.); Wu_lumei@163.com (L.W.); zhenyan252858@163.com (Z.Y.); 17628439383@163.com (L.Z.); y2908363303@126.com (Y.Y.); liangkj_2005@126.com (K.L.); lwx@fafu.edu.cn (W.L.); 2Key Laboratory of Crop Ecology and Molecular Physiology (Fujian Agriculture and Forestry University), Fujian Province University, Fuzhou 350002, China

**Keywords:** inositol polyphosphate 5-phosphatase, soybean, salt tolerance, abscisic acid, BY-2, Arabidopsis, hairy roots

## Abstract

Inositol polyphosphate 5-phosphatases (5PTases) function in inositol signaling by regulating the catabolism of phosphoinositol derivatives. Previous reports showed that 5PTases play a critical role in plant development and stress responses. In this study, we identified a novel 5PTase gene, *Gs5PTase8*, from the salt-tolerance locus of chromosome 3 in wild soybean (*Glycine soja*). *Gs5PTase8* is highly up-regulated under salt treatment. It is localized in the nucleus and plasma membrane with a strong signal in the apoplast. Ectopic expression of *Gs5PTase8* significantly increased salt tolerance in transgenic BY-2 cells, soybean hairy roots and Arabidopsis, suggesting Gs5PTase8 could increase salt tolerance in plants. The overexpression of *Gs5PTase8* significantly enhanced the activities of catalase and ascorbate peroxidase under salt stress. The seeds of *Gs5PTase8*-transgenic Arabidopsis germinated earlier than the wild type under abscisic acid treatment, indicating Gs5PTase8 would alter ABA sensitivity. Besides, transcriptional analyses showed that the stress-responsive genes, *AtRD22*, *AtRD29A* and *AtRD29B*, were induced with a higher level in the *Gs5PTase8*-transgenic Arabidopsis plants than in the wild type under salt stress. These results reveal that Gs5PTase8 play a positive role in salt tolerance and might be a candidate gene for improving soybean adaptation to salt stress.

## 1. Introduction

Cultivated soybean (*Glycine max*) is an important economic crop as a main source of dietary protein and oil. However, its development and agricultural productivity could be severely restricted by salt stress [1,2]. Unfortunately, around 20% irrigated land is salt affected and the salt-affected regions continue to expand [3,4]. Thus, it is necessary to improve salt tolerance in cultivated soybean. On the other hand, wild soybean (*Glycine soja*), the close related species of *Glycine max*, could confer higher tolerance to salinity or other stresses. Wild soybean could supply resistant gene resources for molecular breeding in soybean. It is meaningful to identify new salt-tolerance genes from *Glycine soja* and characterize their molecular mechanism. 

Inositol polyphosphate 5-phosphatases (5PTases) are signal modifying enzymes with the conserved inositol polyphosphate phosphatase catalytic (IPPc) domain, which could hydrolyze the phosphate bond at the 5 position of the inositol ring from inositol phosphate or phosphoinositide [5,6]. Thus, the 5PTases have been suggested to inhibit or terminate inositol signaling, correspondingly altering abscisic acid (ABA) signaling, Ca^2+^ oscillation, and redox homeostasis [7,8,9]. Previous reports have illustrated that 5PTases participate in various processes of development and stress responses in plants [7,10,11,12]. 

The *5PTase* genes are widespread among plants. For example, there are 15 members in *Arabidopsis thaliana*, 21 in rice and 39 in soybean [13]. Until now, the knowledge on the functions of plant 5PTases is still limited, and most is from the model plant Arabidopsis by examining the plants containing gain or loss of function in a certain 5PTase. At5PTase7 and At5PTase9 have been identified to take a crucial role in salt tolerance [14,15]. Mutations in *At5PTase7* or *AtPTase9* would result in increased salt sensitivity, whereas overexpression of *At5PTase7* or *AtPTase9* would enhance salt tolerance in Arabidopsis [14,15]. Both of the *At5ptase7* and *Atptase9* mutants would disturb reactive oxygen species (ROS) production and the expression of salt-responsive genes. Besides, the *Atptase9* mutants also showed reduced Ca^2+^ influx and decreased fluid-phase endocytosis, suggesting At5PTase7 and AtPTase9 have non-redundant roles in the regulation of plant salt tolerance [14,15]. 

On the other hand, GmSAL1, a soybean inositol 1-phosphatases, has been found to be involved in salt and drought response by the hydrolysis of inositol-1,4,5-trisphosphate (InsP_3_) [16], suggesting the inositol signaling would also play a role in stress responses in soybean. The components including 5PTases, involved in inositol signaling, would be good candidates for soybean breeding of salt tolerance as well. However, the function of most 5PTases remains unknown in soybean. 

Recently, a salt-tolerance locus has been identified in chromosome 3 of *Glycine soja* and the salt-tolerant *Glycine max* varieties [17,18,19]. In order to dig out more salt-tolerance genes, we performed post-genomonic analyses on this locus. In this study, a novel soybean *5PTase* gene, *Gs5PTase8*, has been identified from this salt tolerance locus due to its high induction by salinity in the *Glycine soja* accession W05. The analyses of sequences showed that Gs5PTase8 contains the two conserved domain as the other known 5PTases, indicating it is a potential 5PTase. Its effects on plant salt responses were monitored via ectopic expression in BY-2 cells, soybean hairy roots, and *Arabidopsis thaliana*. Its impacts on the activities of antioxidant enzymes, ABA signaling, and stress-responsive gene expression were also determined here. 

## 2. Results

### 2.1. Gs5PTase8 Encoding a Putative Inositol Polyphosphate 5-Phosphatas is Induced by Salt Stress

A novel salt tolerance locus has been identified in the chromosome 3 of soybean previously [17,18,19]. From the transcriptome data of soybean under salt stress [20], we identified a salt-induced gene, *Gs5PTase8* (XM_028369920), in the salt tolerance locus of the *G. soja* accession W05. To verify the induction of *Gs5PTase8* by salinity, the expression analyses were performed in W05 by qRT-PCR (Figure 1A). The results showed that the expression of *Gs5PTase8* was highly up-regulated, especially at 24 h after salt treatment in both leaves and roots, suggesting *Gs5PTase8* would be involved in salt responses. 

*Gs5PTase8* was then cloned from W05. The cDNA sequence contains 1482 bp encoding a peptide of 493 aa. The BLASTP searches showed that its closest homolog in *Arabidopsis thaliana* is *At5PTase8*, so the gene has been named as *Gs5PTase8* (Figure 1B). Its counterpart in cultivated soybean is *Glyma.03G173000*, named as Gm5TPase8 in this study. The analyses of phylogenic relationship were performed with the full-length amino acid sequences of Gs5PTase8, Gm5TPase8, Arabidopsis 5PTases, and other well-known 5PTases from human (*Homo sapiens*), rat (*Rattus norvegicus*), and yeast (*Saccharomyces cerevisiae*) using the neighbor-joining method with 1000 replications of bootstrap tests by MEGA6.0 (Figure 1B). It revealed that Gs5PTase8 and Gm5PTase8 were closer to the Arabidopsis At5PTases without WD-repeats (WD40 domain, cd00200, the GenBank conserved domain database) than those At5PTases with WD-repeats, and also closer to human Type II 5PTase than the other human 5PTases. The sequences analysis indicated that Gs5PTase8 contained the conserved domain I and II of the 5-phoshatase catalytic domains as the other known 5PTases, suggesting *Gs5PTase8* encodes a putative inositol polyphosphate 5-phosphatase (Figure 1C).

### 2.2. Ectopic Expression of Gs5PTase8 Increased Salt Tolerance in Plants

In order to investigate its function in plant salt responses, the effects of Gs5PTase8 were assessed by the utilization of the three transformation systems of BY-2 cell lines, soybean hairy roots and *Arabidopsis thaliana*. The functional analysis in salt responses was performed in BY-2 cells containing the transgenes of *Gs5PTase8* (Figure 2). The expression of the transgenes was confirmed by qRT-PCR. The dead cells were visualized by trypan blue staining. Under normal growth conditions, all the cell lines have around 98% survival rates. Under 100 mM NaCl treatments, all the transgenic lines of *Gs5PTase8* exhibited significantly higher survival rates than the wild type (WT) and the vector-only control line (V).

The soybean cotyledons were infected for hairy root induction by the wild type of *A. rhizogenes* stain K599 (CK), K599 containing the empty binary vector pTEV8 (V) or K599 containing the recombinant construct V7-Gs5PTase8 (Gs5PTase8), respectively. The transformation of hairy roots were confirmed by PCR genotyping (Figure 3A). Nine roots were positive transformants in 10 roots, indicating that the transformation was mainly successful. All the hairy roots grew well with no obvious difference under normal growth condition, whereas the growth of *Gs5PTase8* transformed hairy roots was inhibited at a less level with more fresh weight than that of the controls under 150 mM NaCl (Figure 3). The results revealed that the overexpression of *Gs5PTase8* alleviated salt stress in soybean hairy roots.

Three independent homozygous transgenic Arabidopsis lines of *Gs5PTase8* were obtained for functional analyses under salt stress (Figure 4A). Under normal growth conditions, the seeds of all the lines germinated at a similar rate in 2 days after stratification and there are no significant growth difference between them. Under salt treatment, the germination rates of all the *Gs5PTase8-*transgenic lines were significantly higher than the rates of the wild type (Col-0), especially at 3.5 d after stratification (Figure 4B,C). Moreover, when the 3-day seedlings were treated by 175 mM NaCl for one week, the results showed that the *Gs5PTase8-*transgenic plants had significantly higher survival rates and relative chlorophyll contents comparing to the wild type (Figure 4D–F). The results demonstrated that the ectopic expression of *Gs5PTase8* could increase salt tolerance in Arabidopsis as well, in consisted with its function in BY-2 cells and soybean hairy roots. 

### 2.3. Enhanced Activities of Antioxidant Enzymes in Gs5PTase8-Transgenic Arabidopsis

The evidences suggested that salinity would lead to oxidative stress [3]. To assess the contribution of antioxidant enzymes in reduction of oxidative stress, the activities of catalase (CAT), ascorbate peroxidase (APX), peroxidase (POD), and superoxide dismutase (SOD) were measured in the wild type and the three *Gs5PTase8*-transgenic Arabidopsis lines. As shown in Figure 5, the application of salt stress increased the four ROS scavenging enzymes more or less. Among them, the activities of CAT and APX were significantly higher in the *Gs5PTse8*-transgenic lines than the wild type. It demonstrated that the overexpression of *Gs5PTse8* could enhance the activities of CAT and APX under salt treatment, probably to alleviate the following oxidative stress caused by salinity.

### 2.4. Altered ABA Sensitivity of Gs5PTase8-transgenic Arabidopsis

Since 5PTase enzymes have been observed to be involved in ABA signaling pathway [7], ABA sensitivity were detected with the Arabidopsis plants over-expressing *Gs5PTase8* in seed germination (Figure 6). The germination rates of all the plant lines declined when treated with the increased ABA for around 3.5 d after incubation at 21 °C. In general, the *Gs5PTase8-*transgenic lines showed less sensitive to ABA-elicited inhibition of seed germination compared to the wild type. With exposure to over 5 μM ABA, all the seeds hardly germinated. To further verify the differences on ABA sensitivity between the transgenic lines and the wild type, we used 1 μM ABA to perform the time course tests. It illustrated that the *Gs5PTase8-*transgenic plants germinated at a higher speed compared to the wild type. The ectopic expression of *Gs5PTase8* could impart ABA insensitivity on seed germination, suggesting it might be involved in ABA signaling. 

### 2.5. Expression of the Stress-Responsive Genes in Gs5PTase8-Transgenic Arabidopsis

To investigate whether the ectopic expression of *Gs5PTase8* affects the stress-responsive genes, the expression levels of *AtABA1*, *AtRD22*, *AtRD29A,* and *AtRD29B* were detected in the *Gs5PTase8*-transgenic Arabidopsis lines and the wild type by qRT-PCR (Figure 7). *AtABA1* participates in ABA synthesis, and the other three, *AtRD22*, *AtRD29A,* and *AtRD29B* have been showed to be induced by salt and other abiotic stresses [21,22]. All the tested genes were up-regulated in the first 3 or 6 h post salt treatment, then down-regulated and again up-regulated at 24 h post salt treatment. The expression of *AtABA1* seemed not to be altered significantly by ectopic expression of *Gs5PTase8* in Arabidopsis, indicating Gs5PTase8 might not affect the ABA biosynthesis. On the other hand, the expression levels of the other three stress-responsive genes, *AtRD22*, *AtRD29A,* and *AtRD29B,* were induced more in the transgenic lines compared to the wild type under salt stress, suggesting Gs5PTase8 might take an important role in the regulation of those stress-responsive genes. 

### 2.6. Subcellular Localization of Gs5PTase8

To examine the subcellular localization of Gs5PTase8, the recombinant construct of *Gs5PTase8-GFP* under the control of the 35S promoter were transformed into onion epidermal cells by particle bombardment (Figure 8). The 35S::GFP construct was used as the control. Before plasmolysis, the GFP signal appeared throughout the entire cell except the vacuole for 35S::GFP, whereas it was observed in the nucleus and plasma membrane for 35S::Gs5PTase8-GFP (Appendix A). After plasmolysis, the signals the Gs5PTase8-GFP protein were observed in the nuclear, plasma membrane, cell wall and especially in the apoplast (probably Hechitan strands), compared to the GFP alone protein. It revealed that Gs5PTase8 is expressed in the nucleus and plasma membrane. Besides, Gs5PTase8 targets to apoplast particularly, suggesting it might function in apoplastic pathways.

## 3. Discussion

Inositol polyphosphate 5-phosphatases (5PTases) are one of the key components in inositol pathway [8,9]. Previous studies suggested that they could play an important roles in plant development and adaption to environmental stresses [10,11]. They have been identified to be widely distributed in all kingdoms of organisms [13]. However, only a few plant 5PTases have been investigated, and most are from the model plant, *Arabidopsis thaliana*. In this study, a novel *Gs5PTase8* was cloned from *Glycine soja* and characterized for its function in salt responses. The sequence analysis illustrated that Gs5PTase8 contains the two conserved domain of the reported 5PTases (Figure 1), suggesting its potential function as an enzyme to hydrolyze the phosphate at the 5’ site of the inositol ring. 

Based on the analyses of gene expression, *Gs5PTase8* is highly induced in salt-tolerant accession W05. Moreover, the further functional analyses showed that ectopic expression of *Gs5PTase8* could also confer salt tolerance in transgenic BY-2 cells (Figure 2), soybean hairy roots (Figure 3) and Arabidopsis (Figure 4). The activities of two antioxidant enzymes, CAT and APX, were increased in the *Gs5PTase8*-transgenic Arabidopsis lines under salt treatment (Figure 5), suggesting *Gs5PTase8* might relieve the oxidative damage by salt stress and thus increase salt tolerance. All these revealed that *Gs5PTase8* would be a positive regulator in salt tolerance, which might be a good candidate for soybean breeding. 

5PTases have been found to affect ABA signaling pathway, which could play a vital role in plant adaption, especially to abiotic stresses [7,23]. We have determined the effect of Gs5PTase8 on ABA signaling. The results revealed that the overexpression of *Gs5PTase8* in Arabidopsis caused ABA insensitivity on seed germination (Figure 6), which is similar as the previous reports on the mammalian type I 5PTase, At5PTase1, At5PTase2, and At5PTase6. The transgenic Arabidopsis plants expressing the mammalian type I 5PTase exhibited less sensitivity to ABA treatment and increased drought tolerance comparing to the wild type [24]. At5PTase1 and At5PTase2 could alter ABA signaling via the hydrolysis of inositol-1,4,5-trisphosphate (InsP_3_) [25,26]. Their mutants showed a ABA hypersensitivity phenotype with faster seed germination and longer hypocotyls when grown in dark [27]. Similarly, the *cvp2* mutant (deficient in *At5PTase6*), with open vein network in leaf, also showed an increased sensitivity to ABA in seed germination [28]. But mutation in *At5PTase13* caused ABA insensitivity [29,30]. Thus, it suggested that Gs5PTase8 might also impact ABA signaling for its function. 

Among the reported Arabidopsis 5PTases, At5PTase7, and At5PTase9 have been characterized as a positive regulator in salt tolerance [14,15]. The overexpression of *At5PTase7* or *At5PTase9* displayed enhanced salt induction of the salt-induced marker genes, *AtRD22* and *AtRD29A* [14,15]. We monitored the expression levels of three known stress-marker genes of *AtRD22*, *AtRD29A,* and *AtRD29B* [31] under salt stress. The stress-marker genes could be up-regulated under abiotic stress or ABA treatment. Our observations showed that the expression of *AtRD22*, *AtRD29A,* and *AtRD29B* exhibited increased salt induction in *Gs5PTase8*-transgenic lines compared to the wild type as well (Figure 7). The subcellular localization analysis demonstrated that the fluorescent signal of Gs5PTase8-GFP could be detected in the nucleus. Gs5PTase8 might be involved in the regulation of stress-responsive gene expression by itself or via the ABA signaling pathway.

Except for the nuclear localization, Gs5PTase8-GFP was detected in plasma membrane with a relatively strong signal in the apoplast as well (Figure 8). The apoplast is the portion outside the cell membrane, including the cell wall and the intercellular space, with the function as a barrier and a linker between the environment and the protoplast [32]. Due to its apoplastic localization, we supposed that Gs5PTase8 might hydrolyze the membrane-bound phosphoinositide and be involved in the lipid signaling. However, there are no exploration on the substrates of Gs5PTase8 and its closest Arabidopsis homolog, At5PTase8, yet. Previous reports showed that various components exist in the apoplast, including ions, reactive oxygen species (ROS), and secreted proteins [32]. The apoplastic ROS and the scavenging enzymes have been identified to provide the signals for stress responses [33]. At5PTase7 and At5PTase9 have been identified to increase the production of ROS in the initial period of salt treatment, thus probably triggering the scavenging pathway to tolerate salt stress [14,15]. We have detected the activities of four antioxidant enzymes, CAT, APX, POD, and SOD at 24 h after salt treatment. Among them, CAT and APX were activated with a significant higher level in the *Gs5PTase8*-transgenic Arabidopsis plants than the wild type under salt treatment (Figure 5), suggesting that Gs5PTase8 would regulate ROS signaling probably in the apoplast under salt stress. 

In conclusion, we identified a novel salt-induced Inositol polyphosphate 5-phosphatase gene, *Gs5PTase8*, in soybean and characterized its function in salt responses. The gain-of-function tests demonstrated that Gs5PTase8 conferred salt tolerance in transgenic BY-2 cells, soybean hairy roots and Arabidopsis. Overexpression of *Gs5PTase8* increased the activity of CAT and APX, altered the ABA signaling pathway and up-regulated the expression of salt-responsive genes under salt stress, which might result in enhanced salt tolerance. Gs5PTase8 is localized not only inside the protoplast, but also the apoplast. This study provided a novel gene for improving salt tolerance in soybean and other plants. However, further investigation would be performed to reveal its substrates and its comprehensive roles in the ABA signaling and apoplastic pathways. 

## 4. Materials and Methods

### 4.1. Plant Materials and Growth Conditions

The *Glycine soja* accession W05 (variety name: Mengjin1, salt tolerant) and *Glycine max* accession C08 (variety name: Union, salt sensitive) supplied by Dr. Honming Lam’s lab, Chinese University of HongKong, were used for gene cloning and expression analyses in this work [17,34,35]. After soybean seeds were germinated on vermiculite for one week, the seedlings were transplanted to a hydroponic system of half-strength Hoagland’s solution in a greenhouse [36]. 

Tobacco BY-2 cells [37] and *Arabidopsis thaliana* (ecotype: Columbia-0) were employed to obtain the transgenic lines for functional analyses. BY-2 cells were cultured in Murashige and Skoog (MS) media (Hopebio, Qingdao, China) supplemented with 0.025% KH_2_PO_4_ and 3% sucrose at 25 °C in dark. Arabidopsis plants were grown on 1/2 MS medium or in hydroponic system of 1/10 MS solution under a 16 h light / 8 h dark cycle at 21 °C and 70%–80% humidity. The hydroponic system was used only for the gene expression analyses on Arabidopsis under salt treatment. And the plants were grown on 1/2 MS medium for all the other experiments.

### 4.2. Gene Expression Detected by qRT-PCR

For soybean, when the first trifoliate of the soybean seedling opened, the nutrient solution was replaced by the fresh 1/2 Hoagland’s solution containing 0.9% NaCl (*w*/*v*) [17,20,38]. To extract the total RNA, the leaves and roots were collected and ground in liquid nitrogen separately at 0 h, 4 h, 24 h, 48 h, and 72 h after the start of salt treatments [20,38,39]. For Arabidopsis, three-week-old seedlings grown in the hydroponic system were transferred to the fresh 1/10 MS solution supplemented with 100 mM NaCl. The sampling time points were 0 h, 1 h, 3 h, 6 h, 12 h, and 24 h after the start of salt treatments. Three individual plants under the same treatment were pooled into one biological sample.

The total RNA was extracted using RNAiso PLUS reagent (TaKaRa Biotechnology Co. Ltd. Dalian, China) and the cDNA was synthesized using PrimeScript^TM^RT reagent Kit with gDNA Eraser (TAKARA, Dalian, China) according to the manufacturer’s instructions. Quantitative reverse-transcription PCR (qRT-PCR) was performed using SYBR Premix Ex Taq^TM^ II (Tli RNaseH Plus) (TAKARA, Dalian, China) with the CFX96 Touch^TM^ Real-Time Detection System (Bio-Rad, Hercules, CA, USA). *GmELF1b*, *AtACT2* and *NtL25* were used as internal reference genes, respectively for soybean, Arabidopsis and tobacco BY-2 cells [14,40,41]. The gene specific primers are listed in Appendix A. All the reactions were performed with three replicas. Relative gene expression was quantified using the 2^−ΔΔ*C*t^ method [42]. 

### 4.3. Molecular Cloning and Sequence Analysis of Gs5PTase8

First-strand cDNAs were synthesized with the RNA extracted from the soybean treated by salt stress, using a PrimeScript^TM^ II 1st strand cDNA Synthesis Kit (TaKaRa, Dalian, China) following the manufacturer’s protocol. Phusion High-Fidelity DNA polymerase (ThermoFisher Scientific, USA) was employed for amplifying the coding sequences of *Gs5PTase8* with the gene specific primers (Appendix A) [35]. The thermocycling conditions for PCR were as follows: 98 °C 30 s; 35 cycles of 98 °C 10 s, 58 °C 30 s and 72 °C 60 s; 72 °C 10 min and 12 °C for hold. The amplicons with expected size were gel-purified and cloned to the pBluescript II KS (+) vector between the sites of *XbaI* and *XhoI*. The sequences of cDNAs were sequenced (BGI, Shenzhen) for verification. Then the cDNA was subcloned to the same restriction enzyme sites of the binary vector pTEV8 at the downstream of cauliflower mosaic virus 35S promoter [43]. The recombinant construct was named as V7-Gs5PTase8. After sequence verification, they were transformed into *Agrobacterium rhizogenes* strain K599, *Agrobacterium tumefaciens* strain GV3101 or LBA4404 respectively by a freeze-thaw method [44].

The BLASTP searches were performed to find the homologous sequences (https://blast.ncbi.nlm.nih.gov/Blast.cgi). Multiple sequence alignment was performed with the full length sequences of protein using the CLUSTALX2.1 program [45]. The phylogenetic tree was constructed using MEGA version 6 by the neighbor-joining method with a bootstrap value of 1000 repeats [46].

### 4.4. Cell Viability Assay of Transgenic BY-2 Cells

Tobacco (*Nitcotiana tabacum*) BY-2 cells [37] were transformed with *A. tumefaciens* stain LBA4404 containing the recombinant plasmid, V7-Gs5PTase8 by a co-cultivation method [47]. The positive transformants were selected on the MS medium containing 50 mg/L kanamycin and the expression of the transgene was confirmed by qRT-PCR.

Four-day-old BY-2 cell suspension cultures were treated with 100 mM NaCl for 20 h. The salt-treated and untreated cells were stained with 0.4% trypan blue (Sigma-Aldrich, St. Louis, MO, USA) for 5 min, and then were observed under a light microscope (Nikon Eclipse 80i, Tokyo, Japan). The images were captured by a CCD camera. A total of 200–300 cells were counted for each sample. Three replicates were performed for each sample.

### 4.5. Assays of Hairy Roots of Transformed Soybean Cotyledons under Salt Stress

Hairy root transformation was performed as previously described with minor modifications [48]. The wild type of *A. rhizogenes* stain K599 (CK), K599 containing the empty vector (V), and K599 containing the recombinant construct V7-Gs5PTase8 (Gs5PTase8) were employed to transform cotyledons of any salt-sensitive soybean accessions (such as C08 and Williams 82) for hairy root induction. *A. rhizogenes* stains were grown in lysogeny broth (LB) medium containing 50 mg/L kanamycin and 200 μM acetosyringone at 28 °C for 16 h. The soybean seeds were germinated with water at room temperature for 2–4 days. The cotyledons were cut off by a scalpel and scored by a syringe with needle. The wounds were infected with *A. rhizogenes* K599 for 15 min and then transferred to moist filter paper for co-cultivation in the dark at 25 °C for 5 days. Subsequently, the infected cotyledons were transferred to a growth chamber under 12 h light/12 h dark cycle at 25 °C. When the hairy roots were sprout, the infected cotyledons with the hairy roots of similar lengths (around 1 cm) were selected and treated with 1/2 Hoagland solution with or without 150 mM NaCl for 10 days. The hairy roots were photographed by a single lens reflex camera D7100 (Nikon Imaging (China) Sales Co., Shanghai, China) and the fresh weight of roots was measured. DNA was extracted respectively from each hairy root and the PCR reaction was performed with the *NPTII* primers (kanamycin selection marker of the binary vector pTEV8) to verify the positive transformation. The successful transformants of the empty vector and the recombinant construct would have a PCR product of 400 bps. The fresh weight data of those negative transformants were removed.

### 4.6. Generation of Transgenic Arabidopsis Lines

*A. thaliana* (ecotype: Col-0) plants were transformed by *A. tumefaciens* stain GV3101 containing the recombinant construct V7-Gs5PTase8 using the floral dip method [49]. The transgenic seeds were screened on MS medium with 50 mg/L kanamycin for positive transformants and at least 3 independent T_1_ lines were selected out. The homozygous T_2_ transgenic lines were obtained by selecting on MS medium with kanamycin and their progenies were used for further analysis. The expression of *Gs5PTase8* was verified in the transgenic lines by qRT-PCR.

### 4.7. Phenotype Analyses of Transgenic Arabidopsis Lines

For germination analyses, Arabidopsis seeds were sowed on normal 1/2 MS plates or the medium supplemented with 175 mM NaCl concentrations or varied ABA concentrations (0.25 μM, 0.5 μM, 0.75 μM, 1 μM, 2 μM, 5 μM, or 10 μM) after surface-sterilization. The seeds were stratified at 4 °C in dark for 3 d before shifting to the growth chamber of 21 °C. Then, the germinated seeds were counted every 12 h in 6 days. To determine the best concentration for ABA sensitivity assay, around 150 seeds were used for each sample. In the other germination analyses, a total of around –500 seeds were used for each sample from three independent experiments. 

For survival assays, the 3-day seedlings were transferred to the fresh 1/2 MS media with or without 175 mM NaCl for one week. A total of 75 plants were used for each sample from three replicates. The survival rate was calculated as the percentage of the survival plants after salt treatment. The chlorophyll content was extracted from 0.05 g of rosette leaves from the survival plants by being soaked in 1 mL 95% ethanol at 4 °C in dark for 24 h. The absorbance of the extraction was checked at the wavelength of 649 and 665 nm. Total chlorophyll concentration was calculated as the following equation: C_total_ = C_a_ + C_b_ = ((OD_665 nm_ × 13.95−OD_649 nm_ × 6.88) + (OD_649 nm_ × 24.96−OD_665 nm_ × 7.32))/(sample weight) [50]. 

### 4.8. Assays for Antioxidant Enzyme Activities

The 10 d-seedlings were transferred to the 1/2 MS media with or without 150 mM NaCl for 24 h. Briefly, the fresh leaves of 0.1 g were ground in 2 mL of 0.2 M phosphate buffer saline (PBS, pH 7.8) containing 1% polyvinylpyrrolidone (PVP) on ice. The homogenate was centrifuged at 4000 rpm for half an hour at 4 °C and the supernatant was stored at 4 °C for the activity detection of catalase (CAT) [51], ascorbate peroxidase (APX) [52], superoxide dismutase (SOD) [53] and peroxidase (POD) [54]. The CAT activity was determined by monitoring decreases in absorbance at 240 nm due to the decomposition of H_2_O_2_. The APX activity was assessed by the decline of absorbance at 290 nm for the oxidation of ascorbate. The SOD activity was measured by detecting the absorbance at 560 nm for inhibiting the reduction of nitro blue tetrazolium (NBT) in light. The POD activity was assayed by detecting the absorbance at 470 nm due to the oxidation of guaiacol by H_2_O_2_. Each experiment was carried out at least three times. 

### 4.9. Subcellular Localization of Gs5PTase8

The full length coding sequence of *Gs5PTase8* without the stop codon was amplified and cloned into the XbaI and BamHI restriction enzyme sites of the binary vector pTEV8 at the downstream of 35S promoter. The *GFP* coding sequence was cloned behind *Gs5PTase8* between the sites of BamHI and XhoI. The constructs were coated with gold particles and bombarded into the epidermal cells of onion (Bio-Rad PDS-1000/He system, Hercules, CA, USA). The onion sections were imaged at room temperature using a Leica SP8 X inverted confocal microscope with an Argon laser (Leica, Wetzlar, Germany). Green fluorescence protein (GFP) is excited at 488 nm and the emitted light is captured at 510-525nm. The images were captured digitally and processed using the Leica Application Suite Advanced Fluorescence Lite (LAS AF version: 2.6.3 build 8173). Plasmolysis was performed by incubating the samples in 25% sucrose for 5 min.

### 4.10. Statistical Analyses

Significant tests were performed by Statistical Package for Social Sciences (version 17.0, SPSS Inc. Chicago, IL, USA) statistical package. 

## Figures and Tables

**Figure 1 ijms-21-01023-f001:**
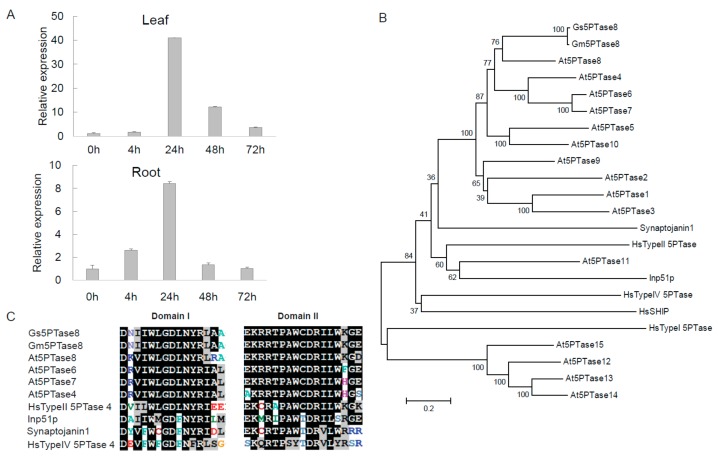
Gs5PTase8, a putative inositol polyphosphate 5-phosphatase, is induced by salt. (**A**) The expression of *Gs5PTase8* in the *Glycine soja* accession W05 under salt treatment. Leaves and roots were collected separately after the salt treatments of 0.9% NaCl (*w*/*v*) at 0 h, 4 h, 24 h, 48 h, and 72 h. The data were normalized to the soybean reference gene *ELF1b*. The value of the samples collected at 0h was set at 1. (**B**) Phylogenic tree of Gs5PTase8 and its homologs using neighbor-joining method with 1000 replications of bootstrap tests by MEGA6.0. The scale bar indicates 0.2 amino acid substitutions per residue. (**C**) Sequence alignments of the two conserved 5-phoshatase catalytic domains of Gs5PTase8 and its homologs. The conserved amino acid residues are shaded in black and the similar residues in gray. Gs5PTase8 is the protein from *Glycine soja* (accession W05), Gm5PTase8 from *Glycine max* (accession Williams 82 / C08), and At5PTases from *Arabidopsis thaliana*.Gs5PTase8: XM_028369920; Gm5PTase8: Glyma.03G173000; At5PTase1: At1G34120; At5PTase2: At4G18010; At5PTase3: At1G71710; At5PTase4: At3G63240; At5PTase5: At5G65090; At5PTase6: At1G05470; At5PTase7: At2G32010; At5PTase8: At2g37440; At5PTase9: At2G01900; At5PTase10: At5g04980; At5PTase11: At1G47510; At5PTase12: At2G43900; At5PTase13: At1G05630; At5PTase14: At2G31830; At5PTase15: At1G65580; HsTypeI 5PTase: X77567 (Hs represents for *Homo sapiens*); HsTypeII 5PTase: M74161; HsTypeIV 5PTase: AF187891; HsSHIP: U57650; Synaptojanin1: U45479 (from *Rattus norvegicus)*; Inp51p: NP_012264 (from *Saccharomyces cerevisiae*).

**Figure 2 ijms-21-01023-f002:**
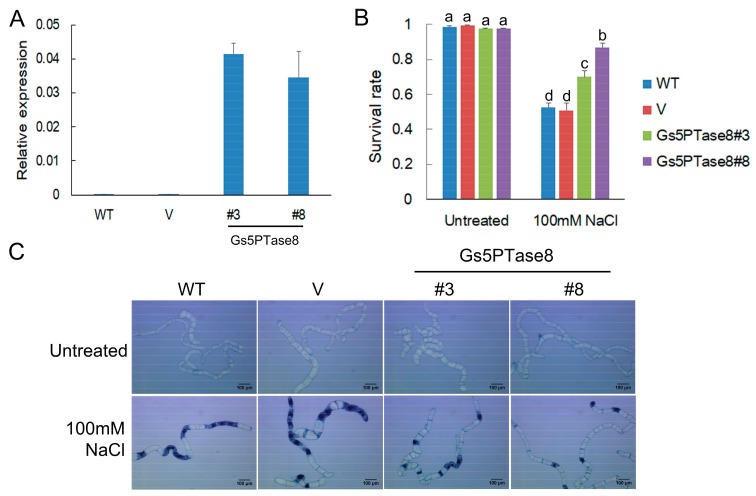
Cell viability assay of BY-2 cells under salt treatment. Four-day-old BY-2 cells cultured in MS medium were used, including the wild type (WT), the vector-only control line (V), two independent *Gs5PTase8* transgenic lines (Gs5PTase8#3 and Gs5PTase8#8). The cells were treated or untreated with 100 mM NaCl for 20 h and the dead cells were visualized by trypan blue staining. (**A**) The expression of transgenes (2^−ΔCt^) were validated by qRT-PCR. The data were normalized to the tobacco reference gene *L25*. (**B**) The quantitative analysis of survival rates. For each sample, around 200~300 cells were counted randomly. Three replicates of each experiment were performed. Values are mean ± SE. The data were analyzed using ANOVA followed by the Duncan’s post hoc test (*p* < 0.05). The significant differences were indicated by different letters above the bars. (**C**) The images of the BY-2 cells by trypan blue staining. Scale bars indicate 100 μm.

**Figure 3 ijms-21-01023-f003:**
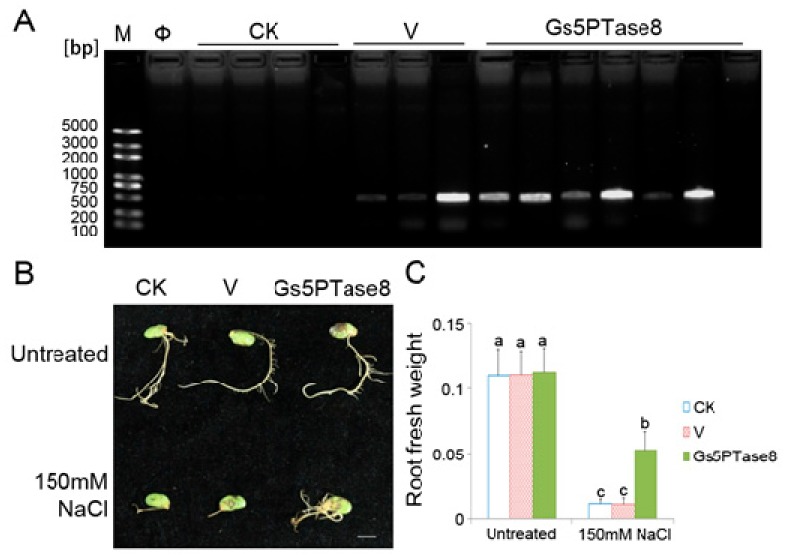
Growth characteristics of soybean hairy roots under salt treatment. The cotyledons of the geminated soybean seeds were infected with the wild type of *A. rhizogenes* stain K599 (CK), K599 containing the empty vector pTEV8 (V) or K599 containing the recombinant construct V7-Gs5PTase8 (Gs5PTase8), respectively. The cotyledons with the hairy roots of around 1 cm were selected for salt treatment. The selected hairy roots were grown with 1/2 Hoagland solution containing 0 mM NaCl and 150 mM NaCl for 10 days. (**A**) PCR verification of the positive transformants. The primers of the *NPTII* gene (kanamycin selection marker) from the binary vector pTEV8 were used for the genotyping PCR. The successful transformants of the empty vector and the recombinant construct would have a PCR product of 400 bps. M represents for DNA marker; Φ, Milli-Q water control of PCR reaction; bp, base pairs. (**B**) Phenotype of hairy roots. Scale bar indicates 1 cm. (**C**) Fresh weight of hairy roots. Sample means with standard errors (SE) were calculated with three replicates each containing 5–6 soybean cotyledons. The significant tests were performed using ANOVA (Duncan’s test, *p* < 0.05). The significant differences were indicated by different letters above the bars.

**Figure 4 ijms-21-01023-f004:**
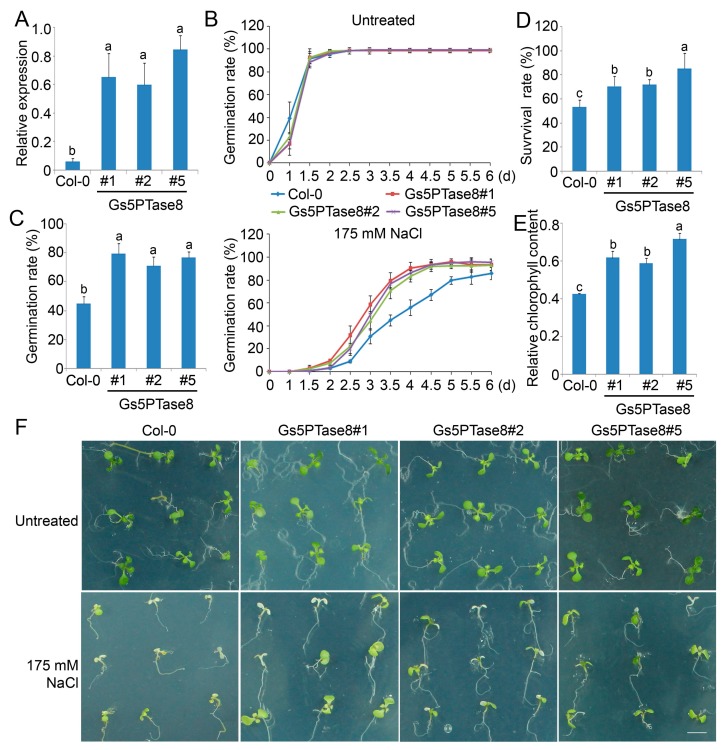
Phenotype of three independent *Gs5PTase8*-transgenic Arabidopsis lines (#1, #2, #5) under salt treatment. (**A**) The expression of *Gs5PTase8* (2^−ΔCt^) were validated in the transgenic lines by qRT-PCR. The data were normalized to the Arabidopsis reference gene *ACT2*. (**B**) Arabidopsis seeds were sowed on the 1/2 MS agar plates with or without 175 mM NaCl and were stratified at 4 °C for 3 d. The germination rates were monitored every 12 h for 6 days after shift to 21 °C. (**C**) The statistical analysis of the germination rates were represented at 3.5 d post stratification. In the germination assays, around 150 seeds were used for each sample. At least three independent experiments were performed. (**D**) The survival rates after salt treatment. Three-day seedlings were treated by 175 mM NaCl for one week. Three replicates were performed with 25 plants for each variant. (**E**) The relative chlorophyll content was shown as the ratio of the chlorophyll content from the untreated plants compared to that from the 175 mM NaCl treated survival plants for each line. (**F**) Phenotype of the plants after the survival assays. Scale bar indicates 1 cm. The data were analyzed using ANOVA by the Duncan’s post hoc test (*p* < 0.05). The significant differences were indicated by different letters above the bars.

**Figure 5 ijms-21-01023-f005:**
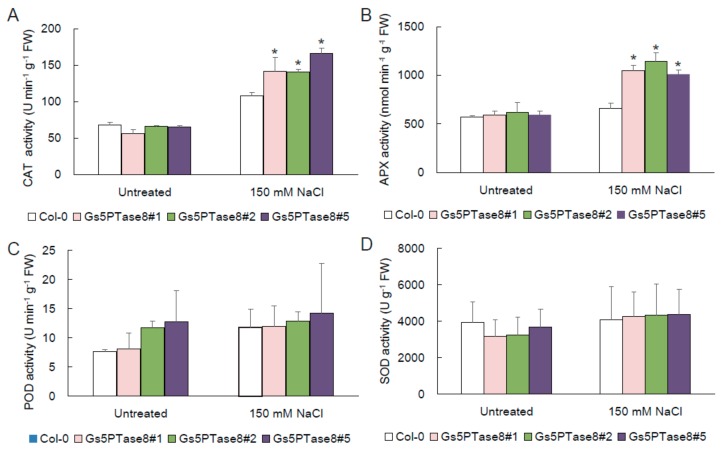
Activities of antioxidant enzymes in *Gs5PTase8*-transgenic Arabidopsis under salt treatment. The ten-day-old Arabidopsis seedlings were treated with or without 150 mM NaCl for 24 h. (**A**) CAT activity, (**B**) APX activity, (**C**) POD activity, and (**D**) SOD activity. The data were analyzed using ANOVA by the Dunnett’s post hoc test. Asterisk indicates a significant difference between the wild type and the transgenic lines (*p* < 0.05).

**Figure 6 ijms-21-01023-f006:**
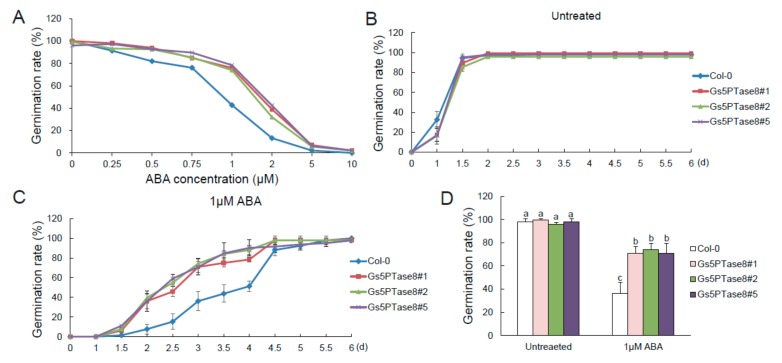
Germination of *Gs5PTase8*-transgenic Arabidopsis under abscisic acid (ABA) treatment. (**A**) The germination rate under a series of ABA treatment. Arabidopsis seeds were sowed on 1/2 MS median with varied ABA (0, 0.25 μM, 0.5 μM, 0.75 μM, 1 μM, 2 μM, 5 μM, and 10 μM), followed by being stratified at 4 °C for 3 d and then shifted to 21 °C. The germination rates were determined at around 3.5 d after seed imbibition for 150 seeds per line. (**B**) The germination rates were monitored every 12 h in 6 days after incubation at 21 °C under normal conditions (untreated). (**C**) The germination rates under 1 μM ABA treatment. (**D**) The statistical analysis of the germination rates at 3 d post stratification were performed by ANOVA with the Duncan’s post hoc test (*p* < 0.05). The significant differences were indicated by different letters above the bars. A total of around 400–500 seeds were used for each sample from three independent experiments.

**Figure 7 ijms-21-01023-f007:**
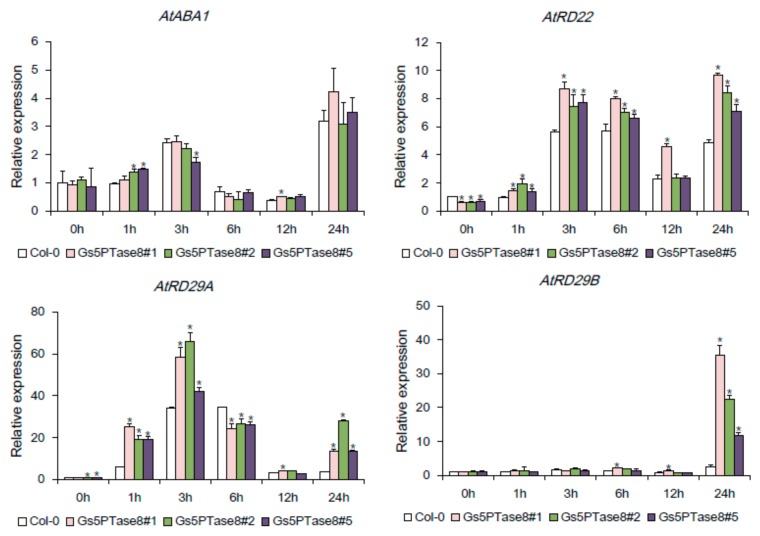
Effects of Gs5PTase8 on the expression levels of the stress-responsive genes. The three-week seedlings were treated by 100 mM NaCl in a hydroponic system for 0 h, 1 h, 3 h, 6 h, 12 h, and 24 h. The expression of *AtABA1*, *AtRD22*, *AtRD29A,* and *AtRD29B* were analyzed by qRT-PCR in the three independent *Gs5PTase8*-transgenic lines (Gs5PTase8 #1, #2, #3) and the wild type (Col-0). The data were normalized to the reference gene *ACT2*. The value of the samples collected from Col-0 at 0h was set at 1. The data were analyzed using ANOVA by the Dunnett’s post hoc test. Asterisk indicates a significant difference between the wild type and the transgenic lines at the same time point (*p* < 0.05).

**Figure 8 ijms-21-01023-f008:**
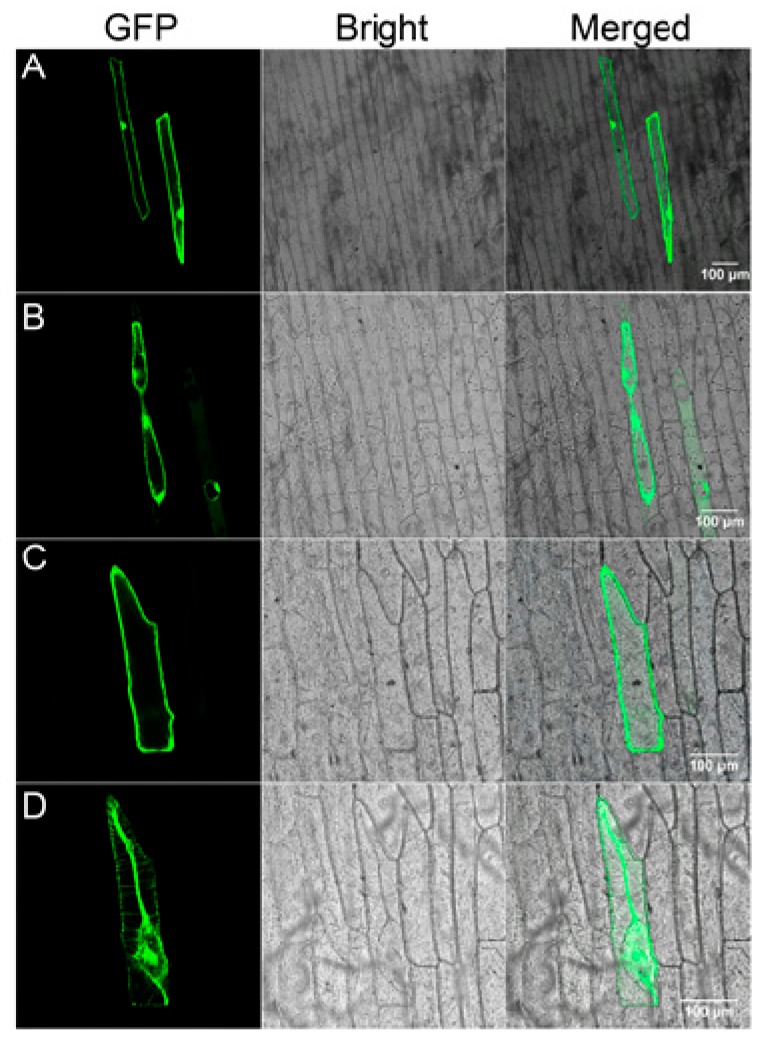
Subcellular localization of Gs5PTase8 in onion epidermal cells. Green fluorescence protein (GFP) (**A, B**) and Gs5PTase8-GFP (**C, D**) driven by CaMV 35S were transiently expressed in onion epidermal cells and observed before (**A, C**) or after (**B, D**) plasmolysis. GFP fluorescence, bright-field and merged images are shown. Scale bars indicate 100 μm.

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
