# Peer review of "Ectopic Expression of Gs5PTase8, a Soybean Inositol Polyphosphate 5-Phosphatase, Enhances Salt Tolerance in Plants"

_ijms, 2020, doi:10.3390/ijms21031023_

Round 1
Reviewer 1 Report
The manuscript focused on Gs5PTase8 shows interesting results usable in genetic engineering and breeding. The main positives of the article consist of the investigation of Gs5PTase8 effect on salt stress tolerance in three different systems (BY2, soybean hairy roots, and stable transformants of Arabidopsis thaliana). In all systems, more independent transformants were used, which supports the results. The results about salt stress tolerance are consistent and they are in agreement with other studies focused on 5PTases. The positive effect of Gs5PTase8 on salt stress tolerance was also supported by the expression of stress-related genes. The phylogenetic tree and domain identification are clear. The link between Gs5PTase8 and abscisic acid, and localization of the protein were also investigated.
The manuscript has some deficiencies. The English language has to be improved as well as many sentences have to be simplified (to be clear). Do not use words like “it is believed” – the results were observed and supported by more experiments. Do not use plural words in the case of “rate”, especially in the description of plot axes. When you write Arabidopsis in text, use the Latin shortcut of species name “A. thaliana” or English name “arabidopsis” (or thale cress). Also, the Latin name Glycine soja (G.soja) should be mentioned in the text more often (especially in the final paragraph of Introduction, in Results and Discussion), partially substituting name wild soybean.
The localization of the protein (chapter 2.6) is incorrect. Firstly, use the same scale of Fig.8A,B as in the case of Fig.7C,D. It should be better to visualize cells in DIC to see better the nucleus. Secondly, GFP itself is localized in the cytosol and nucleus, but Gs5PTase8 is localized in the plasma membrane. It is clearly visible after plasmolysis (Hechtian strands are a good proof of plasma membrane localization). The other proof could be e.g. existence of plasma membrane domains in the protein sequence or some anchor domain. The localization of Gs5PTase8 in the nucleus is visible only in the case of plasmolysis and it can be only artificial caused by plasmolysis and strong expression under 35S promoter. For better evaluation, you could co-localize the protein with some marker of the plasma membrane. The localization could be also in the apoplast, but it should be better investigated.
In Abstract, use whole names of enzymes CAT and APX instead of abbreviations (the same in the case of ABA in keywords).
lines 65-66 – it is not clear if the analysis of Chromosome 3 was made in this study or in studies 17-19. Correct the sentence.
line 78 – link to Figure 1A
line 83 – correct “so we named it” to “the gene has been named”. Add also the link to Fig.1B.
line 84 – correct “named as Gm5PTase8 here” to “named as Gm5PTase8 in this study”
line 86 – which “other well-known 5PTases”? Write at least names of species that you selected.
line 87 – link to Fig.1B
line 88 – describe what WD-repeats are
line 91 – link to Fig1C
line 94 – reduce the description of Fig.1A to be more clear. It is not necessary to write about the calculation of qRT-PCR because it is mentioned in Materials and Methods (the same for legends of other Figures). It should be better to write to which was the relative expression related (or explain it in Materials and Methods). What does it mean 0.9% NaCl? Add (w/v) or express it as concentration.
line 102 – put the accession numbers to parentheses and add names of organisms from which the sequences originated.
line 112 – do not write “first” (“secondly”...). Just start with BY2 results and then continue with hairy roots results and A. thaliana results in new paragraphs.
Fig.2,3,4 – change the organization of figures. The proof of transformations should be the first figure (A). And the description of the successful transformation should be mentioned for the first also in the text in Results.
Fig.2 – (A) figure is about visualization of dead cells, not about visualization of survived cells (description on line 122). The scale bar in (A) is not clearly visible. (B) missing letters of statistics in untreated samples.
line 131 – the description of used variants and vectors is not clear. Correct the sentence.
line 132-135 – the description of used primers for expression analysis and product size could be mentioned only in the figure legend (in a shortened version) and Materials and Methods.
Fig.3 – (A) why you show three Gs5PTase8 transformed cotyledons? Is it somehow connected with (C)? The description of variants with and without vectors is confusing (line 140-141). Simplify the description of cultivation, statistics (add ANOVA), and (C) abbreviation description in the legend.
line 150 – add a link to Fig.4E
line 155-156 – the description of salt treatment is not clear
Fig.4 – (A) the x axes are hardly transparent. (B) it is not clear that plants were salt-stressed (specify it in the legend). (E) missing statistics; why you have an expression also in Col-0?
line 160 – add “three independent transgenic lines (#1, #2, #5) under salt treatment” and delete this at the end of the legend.
line 165-166 – I do not understand the treatment.
line 166 – 25 plants per each variant
line 174 – write the first sentence to be more scientific.
Fig.6 – (A) errors are missing; the x axis should be without µM and the unit should be written in the axis description; the legend is missing (or use one legend for A,B,C figures). (B,C) x axes labels are not fitting with marks. In legend, add ANOVA and the number of plants used for statistics.
Fig.7 – Statistics is missing. Show the plots legend more compact. Add shortly the cultivation conditions as you write in the legends of other figures.
In Discussion, add links to figures. Write the third paragraph to be more clear and discuss more your results with the other studies.
line 313 – add examples of experiments for which you used MS solid medium or hydroponics.
line 317 - Add (w/v) or express % as concentration.
line 394 – write the sentence about plant samples to be clear.
line 405 – add the information that the measurement was performed spectrophotometrically.
The plots in Figures could look similar (similar pattern, lines, color...).
In Abbreviations – you do not use WT in text.
Author Response
The manuscript focused on Gs5PTase8 shows interesting results usable in genetic engineering and breeding. The main positives of the article consist of the investigation of Gs5PTase8 effect on salt stress tolerance in three different systems (BY2, soybean hairy roots, and stable transformants of Arabidopsis thaliana). In all systems, more independent transformants were used, which supports the results. The results about salt stress tolerance are consistent and they are in agreement with other studies focused on 5PTases. The positive effect of Gs5PTase8 on salt stress tolerance was also supported by the expression of stress-related genes. The phylogenetic tree and domain identification are clear. The link between Gs5PTase8 and abscisic acid, and localization of the protein were also investigated.
The manuscript has some deficiencies. The English language has to be improved as well as many sentences have to be simplified (to be clear). Do not use words like “it is believed” – the results were observed and supported by more experiments. Do not use plural words in the case of “rate”, especially in the description of plot axes. When you write Arabidopsis in text, use the Latin shortcut of species name “A. thaliana” or English name “arabidopsis” (or thale cress). Also, the Latin name Glycine soja (G.soja) should be mentioned in the text more often (especially in the final paragraph of Introduction, in Results and Discussion), partially substituting name wild soybean.
Thanks for your comments and suggestions. We have revised these points.
The localization of the protein (chapter 2.6) is incorrect. Firstly, use the same scale of Fig.8A,B as in the case of Fig.7C,D. It should be better to visualize cells in DIC to see better the nucleus. Secondly, GFP itself is localized in the cytosol and nucleus, but Gs5PTase8 is localized in the plasma membrane. It is clearly visible after plasmolysis (Hechtian strands are a good proof of plasma membrane localization). The other proof could be e.g. existence of plasma membrane domains in the protein sequence or some anchor domain. The localization of Gs5PTase8 in the nucleus is visible only in the case of plasmolysis and it can be only artificial caused by plasmolysis and strong expression under 35S promoter. For better evaluation, you could co-localize the protein with some marker of the plasma membrane. The localization could be also in the apoplast, but it should be better investigated.
We revised the scale for Fig.8. We used two online tools to predict the subcellular localization. MultiLoc2: Predicted Location: cytoplasmic: 0.48 nuclear: 0.41 chloroplast: 0.07 mitochondrial: 0.03 secretory pathway: 0.01; WoLF PSORT: queryProtein details nucl: 7, mito: 3, cyto: 2, chlo: 1, cysk: 1. Our experimental results showed that the signal of Gs5PTas8-GFP could be found in nucleus and cytoplasm. Unexpectedly, after plasmolysis, the signals were also clearly observed in the space between the plasma membrane and the cell wall, supposed to be Hechtian strands. We have observed at least 5 transformed onion cells for Gs5TPase8-GFP and the green signal could be detected in the nucleus and heschtian strands, though sometime it need to be found at different layers of Z-axis. We used DAPI to stain the nucleus and it is in the nucleus. We also transformed BY-2 cells for detection and also found the GFP signals existed in the nucleus. Somehow, it is obscure about its cellular localization and its function. Please see the following photos. (see the attachment)
In Abstract, use whole names of enzymes CAT and APX instead of abbreviations (the same in the case of ABA in keywords).
Done.
lines 65-66 – it is not clear if the analysis of Chromosome 3 was made in this study or in studies 17-19. Correct the sentence.
The previous studies (17-19) have identified a salt-tolerance locus on chromosome 3 in Glycine soja or the salt-tolerant cultivars. Our study focused on Gs5Ptase8, which is located in this salt-determinant locus. We rewrote the sentence.
line 78 – link to Figure 1A
Done.
line 83 – correct “so we named it” to “the gene has been named”. Add also the link to Fig.1B.
Done.
line 84 – correct “named as Gm5PTase8 here” to “named as Gm5PTase8 in this study”
Done.
line 86 – which “other well-known 5PTases”? Write at least names of species that you selected.
Those well-studied 5PTases are from human (Homo sapiens), rat (Rattus norvegicus) and yeast (Saccharomyces cerevisiae). We have added the information in the manu.
line 87 – link to Fig.1B
Done.
line 88 – describe what WD-repeats are
WD40 domain, cd00200, from the GenBank conserved domain database. We have added the information in the manu.
line 91 – link to Fig1C
Done.
line 94 – reduce the description of Fig.1A to be more clear. It is not necessary to write about the calculation of qRT-PCR because it is mentioned in Materials and Methods (the same for legends of other Figures). It should be better to write to which was the relative expression related (or explain it in Materials and Methods). What does it mean 0.9% NaCl? Add (w/v) or express it as concentration.
We renewed the description for all the qRT-PCRs. We added (w/v) to “0.9% NaCl”.
line 102 – put the accession numbers to parentheses and add names of organisms from which the sequences originated.
Done.
line 112 – do not write “first” (“secondly”...). Just start with BY2 results and then continue with hairy roots results and A. thaliana results in new paragraphs.
Done.
Fig.2,3,4 – change the organization of figures. The proof of transformations should be the first figure (A). And the description of the successful transformation should be mentioned for the first also in the text in Results.
Done.
Fig.2 – (A) figure is about visualization of dead cells, not about visualization of survived cells (description on line 122). The scale bar in (A) is not clearly visible. (B) missing letters of statistics in untreated samples.
Done.
line 131 – the description of used variants and vectors is not clear. Correct the sentence.
We revised the description of used variants and vectors as: the wild type of A. rhizogenes stain K599 (CK), K599 containing the empty binary vector pTEV8 (V) or K599 containing the recombinant construct V7-Gs5PTase8 (Gs5PTase8).
line 132-135 – the description of used primers for expression analysis and product size could be mentioned only in the figure legend (in a shortened version) and Materials and Methods.
Done.
Fig.3 – (A) why you show three Gs5PTase8 transformed cotyledons? Is it somehow connected with (C)? The description of variants with and without vectors is confusing (line 140-141). Simplify the description of cultivation, statistics (add ANOVA), and (C) abbreviation description in the legend.
The bar chart of the root fresh weight was about the data collected from 15 ~ 20 roots for each sample. We remained one Gs5PTase8 transformed cotyledon as a representative. The other revision were also been made according to the comments.
line 150 – add a link to Fig.4E
Done.
line 155-156 – the description of salt treatment is not clear
We rewrote it as “When 3-day seedlings were treated by 175 mM NaCl for one week, the results showed that the Gs5PTase8-transgenic plants had significantly higher survival rates and relative chlorophyll contents comparing to the wild type.”
Fig.4 – (A) the x axes are hardly transparent. (B) it is not clear that plants were salt-stressed (specify it in the legend). (E) missing statistics; why you have an expression also in Col-0?
We have revised the Figure 4. We detected very low expression in Col-0 by qRT-PCR, probably due to its homologous gene in thale cress, which would produce unspecific PCR products of very low amount.
line 160 – add “three independent transgenic lines (#1, #2, #5) under salt treatment” and delete this at the end of the legend.
Done.
line 165-166 – I do not understand the treatment.
Three-day seedlings were treated by 175 mM NaCl for one week. The survival rates were determined. We revised the discription.
line 166 – 25 plants per each variant
Done.
line 174 – write the first sentence to be more scientific.
Revised.
Fig.6 – (A) errors are missing; the x axis should be without µM and the unit should be written in the axis description; the legend is missing (or use one legend for A,B,C figures). (B,C) x axes labels are not fitting with marks. In legend, add ANOVA and the number of plants used for statistics.
Fig.6 – (A) is the results from the experiment to determine which ABA concentration is best for the treatment. We tested around 150 seeds for each sample at one time without grouping, and thus no error bar for it. The other germination experiments were done with around 150 seeds for three times and the data of them could have errors. See the details in the method. For the left, we adjusted the figure and legends according to the comments.
Fig.7 – Statistics is missing. Show the plots legend more compact. Add shortly the cultivation conditions as you write in the legends of other figures.
Revised.
In Discussion, add links to figures. Write the third paragraph to be more clear and discuss more your results with the other studies.
Revised.
line 313 – add examples of experiments for which you used MS solid medium or hydroponics.
We added the information there. We have also mentioned it in the context for each related experiment.
line 317 - Add (w/v) or express % as concentration.
Done.
line 394 – write the sentence about plant samples to be clear.
Done.
line 405 – add the information that the measurement was performed spectrophotometrically.
Done.
The plots in Figures could look similar (similar pattern, lines, color...).
Revised.
In Abbreviations – you do not use WT in text.
We used it in Fig.2 and the related text.

Reviewer 2 Report
Review of the manuscript
[IJMS] Manuscript ID: ijms-692746
Title: Ectopic expression of Gs5PTase8, a soybean inositol polyphosphate 5-phosphatase, enhances salt tolerance in plants
Presented manuscript described the identified a novel 5PTase gene in wild soybean. It was shown that Gs5PTase8 is highly up-regulated under salt treatment. Analysis on the onion epidermis has been shown that it is localized throughout the cytoplasm and cell wall. Here, I would suggest to check if it is present in Hechtian strands.
Authors showed that ectopic expression of Gs5PTase8 increase salt tolerance in transgenic BY-2 cells, soybean hairy roots and Arabidopsis. Moreover, Authors showed that the overexpression of Gs5PTase8 would significantly enhance the activities of CAT and APX under salt tress and that Gs5PTase8 alter ABA sensitivity. Important for this studies was the transcriptional analyses which showed that the stress-responsive genes, AtRD22, AtRD29A and AtRD29B, were induced with a higher level in the Gs5PTase8-transgenic Arabidopsis plants than in the wild type under salt stress.
Manuscript is supplemented with the 8 tables in the main text and 1 as a supplementary. Introduction is adequate to the problem, results are well documented, discussion is well written, and materials and methods are accurate and well described.
In my opinion this manuscript should be published in The International Journal of Molecular Sciences after minor revision (see below). The detailed comments are in the PDF file.
1/keywords – should be supplemented (see comment in PDF file)
2/line 62 – word “take” may be better would be “play” (the same concerns line 240)
3/line 374 – more details about the microscope/stereomicroscope, the company.. are needed
4/fig. 8 - I would suggest to considering whether on D there are Hechtian strands? It would be interesting to check it.

Author Response
Review of the manuscript
[IJMS] Manuscript ID: ijms-692746
Title: Ectopic expression of Gs5PTase8, a soybean inositol polyphosphate 5-phosphatase, enhances salt tolerance in plants
Presented manuscript described the identified a novel 5PTase gene in wild soybean. It was shown that Gs5PTase8 is highly up-regulated under salt treatment. Analysis on the onion epidermis has been shown that it is localized throughout the cytoplasm and cell wall. Here, I would suggest to check if it is present in Hechtian strands.
Authors showed that ectopic expression of Gs5PTase8 increase salt tolerance in transgenic BY-2 cells, soybean hairy roots and Arabidopsis. Moreover, Authors showed that the overexpression of Gs5PTase8 would significantly enhance the activities of CAT and APX under salt tress and that Gs5PTase8 alter ABA sensitivity. Important for this studies was the transcriptional analyses which showed that the stress-responsive genes, AtRD22, AtRD29A and AtRD29B, were induced with a higher level in the Gs5PTase8-transgenic Arabidopsis plants than in the wild type under salt stress.
Manuscript is supplemented with the 8 tables in the main text and 1 as a supplementary. Introduction is adequate to the problem, results are well documented, discussion is well written, and materials and methods are accurate and well described.
In my opinion this manuscript should be published in The International Journal of Molecular Sciences after minor revision (see below). The detailed comments are in the PDF file.
Thanks for your comments and suggestions. We have revised all the points according to your comments.
1/keywords – should be supplemented (see comment in PDF file)
We added the others.
2/line 62 – word “take” may be better would be “play” (the same concerns line 240)
Done.
3/line 374 – more details about the microscope/stereomicroscope, the company.. are needed
The hairy roots were photographed by a single lens reflex camera (D7100, Nikon).
4/fig. 8 - I would suggest to considering whether on D there are Hechtian strands? It would be interesting to check it.
Thanks for your suggestions. I think it would be Hechtian strands with a big chance. We would like to check it.
Reviewer 3 Report
This manuscript reports on the characterization of a Glycine soja gene (encoding inositol-P phosphatase) performed by overexpression in heterologous systems. The rationale of the study is well explained and the experiments were performed with sound methods. Results are correctly commented (with one exception, see below). The discussion is sound although some points are not completely addressed (see below).
The (minor) problems I see in this paper are:
it is concluded that the gene is a marker of salt stress tolerance and correspondingly it was isolated in a salt-stress tolerant species. However the gene is also present in Glycine max which is not salt-tolerant, but I could find no further comment on this gene. What are the differences between the Gs and Gm genes? Is the Gm gene also upregulated upon salt stress? Would introgression of the Gs gene into Gm improve salt tolerance of Gm? results suggest that Gs overexpressors are less sensitive to ABA while they show higher activity/expression of enzymes/gene that are activated by An explanation for this contradictory observation should be provided. the cell localization experiments do not convince me. Although I am not a specialist in the filed, it seems clear that the GFP fusion is mostly present on membranes (and not in the apoplast: on plasmolysis the signal remains close to what probably are the collapsed membranes, where inositides are expected to be) while the cell walls show no signal. So I don’t agree with the relevance given to “apoplastic signaling”, and conversely antioxidant enzymes are mostly localised in the cytoplasm and not in the apoplast. final point is english use ( see e.g. “in consisted with” line 157, “The activates of” line 289)Author Response
This manuscript reports on the characterization of a Glycine soja gene (encoding inositol-P phosphatase) performed by overexpression in heterologous systems. The rationale of the study is well explained and the experiments were performed with sound methods. Results are correctly commented (with one exception, see below). The discussion is sound although some points are not completely addressed (see below).
The (minor) problems I see in this paper are:
it is concluded that the gene is a marker of salt stress tolerance and correspondingly it was isolated in a salt-stress tolerant species. However the gene is also present in Glycine max which is not salt-tolerant, but I could find no further comment on this gene. What are the differences between the Gs and Gm genes? Is the Gm gene also upregulated upon salt stress? Would introgression of the Gs gene into Gm improve salt tolerance of Gm?
Thanks for your comments and suggestions. The gene is indeed present in Glycine max. The comparison analysis showed that there were 8 amino-acid substitutions (K10T, I38T, P195R, L198S, A442V., L448M, I487L, I490V) between Gm5PTase and Gs5PTase8, none of which was located in the conserved domain I and II. Gm5PTase8 was also induced upon salt stress, but at a lower level. The differences at the transcriptional level might be the reason for the differences of salt tolerance between C08 and W05. I think the introgression of some Glycine soja genes could improve salt tolerance of Glycine max.
results suggest that Gs overexpressors are less sensitive to ABA while they show higher activity/expression of enzymes/gene that are activated by An explanation for this contradictory observation should be provided.
The induction of antioxidant enzymes could be caused by various pathways including ABA. Here we found that the overexpression of Gs5PTase8 caused ABA insensitivity on seed germination in Arabidopsis. The 5PTase proteins are supposed to function in the inositol pathway directly, and then take an impact on the ABA-dependent or ABA-independent pathway or both. It would function as the similar mode of the ectopic expression of the Type 1 inositol 5-phosphatase in Arabidopsis (Pereea, 2008, Plant cell). But the detailed mechanism is still needed to be explored.
the cell localization experiments do not convince me. Although I am not a specialist in the field, it seems clear that the GFP fusion is mostly present on membranes (and not in the apoplast: on plasmolysis the signal remains close to what probably are the collapsed membranes, where inositides are expected to be) while the cell walls show no signal. So I don’t agree with the relevance given to “apoplastic signaling”, and conversely antioxidant enzymes are mostly localised in the cytoplasm and not in the apoplast.
After plasmolysis, there were signals around the cell wall and in the connected threads between the cell wall and the plasma membrane, it would probably be in Hechtian strands. Antioxidant enzymes are mostly localized in the cytoplasm. But ROS also existed in the apoplast for signal transduction (see ref 30, Zhang, 2009, Plant Physiol.).
final point is english use ( see e.g. “in consisted with” line 157, “The activates of” line 289)
Done.
Round 2
Reviewer 1 Report
I checked your corrections and I do not agree with the Fig.8 re-scaling. You just corrected the scale bar, but the scale of A and B photos are still on the same scale as before.
I read the attachment and I still do not agree with the localization of the protein. You are right that the localization is also in the nucleus, which is better visualized in the other z-axis section. This photo (the second one with plasmolysis from the attachment) should be included e.g. in the supplementary figures. However, the localization seems to be in the plasma membrane, not in the cytoplasm. The first photo from the attachment also supports this theory. The photo is from another z-section in the case of the confocal microscopy, but the visualization from the light microscopy is not shifted to the same z-section. So, you cannot say that the localization is in the cytoplasm, and if you use these photos for the analysis of the localization by software, you could obtain false results. If the localization was in the cytoplasm, you would have seen on this photo also organelles. But they are not visible because the localization is in the plasma membrane (or very close to the membrane). The localizations in apoplast and vacuole are problematic because of low pH. The GFP protein does not function at the very low pH. It should be fine in the subsequent study to make the fusion with another fluorescence dye resistant to pH. The analysis of the protein localization in silico predicts that the Gs5PTase8 is mainly localized outside the plasma membrane, with a small probability to be also in the plasma membrane. The article Kaye et al. 2011 Plant Physiology is about the localization of At5PTase7 and the localization looks similar to your protein. I suggest changing the description of the protein localization within the whole manuscript. Or if you still insist on your description, add also the results from MultiLoc2 and WoLF PSORT into the manuscript.
The correction of the errors and bad sentences was improved. However, the English language still has to be corrected (e.g. there are grammatical errors concerning the third person singular, in Abstract specifically and throughout the paper, conditional tense is used, which suggests that the authors have not proved the facts and they are only suggesting).
line 26 - the abbreviation ABA in Abstract is not explained
Fig.5, Fig.7 - in the legend, the name of the statistic is missing
Fig.6 - in D, the statistic is missing
The English language still has to be corrected because of some elementary grammar mistakes.
Author Response
I checked your corrections and I do not agree with the Fig.8 re-scaling. You just corrected the scale bar, but the scale of A and B photos are still on the same scale as before.
Thanks for your suggestions. I made a mistake to take the universal scale. The original magnification is different for each photo. I checked the magnification and corrected it.
I read the attachment and I still do not agree with the localization of the protein. You are right that the localization is also in the nucleus, which is better visualized in the other z-axis section. This photo (the second one with plasmolysis from the attachment) should be included e.g. in the supplementary figures. However, the localization seems to be in the plasma membrane, not in the cytoplasm. The first photo from the attachment also supports this theory. The photo is from another z-section in the case of the confocal microscopy, but the visualization from the light microscopy is not shifted to the same z-section. So, you cannot say that the localization is in the cytoplasm, and if you use these photos for the analysis of the localization by software, you could obtain false results. If the localization was in the cytoplasm, you would have seen on this photo also organelles. But they are not visible because the localization is in the plasma membrane (or very close to the membrane). The localizations in apoplast and vacuole are problematic because of low pH. The GFP protein does not function at the very low pH. It should be fine in the subsequent study to make the fusion with another fluorescence dye resistant to pH. The analysis of the protein localization in silico predicts that the Gs5PTase8 is mainly localized outside the plasma membrane, with a small probability to be also in the plasma membrane. The article Kaye et al. 2011 Plant Physiology is about the localization of At5PTase7 and the localization looks similar to your protein. I suggest changing the description of the protein localization within the whole manuscript. Or if you still insist on your description, add also the results from MultiLoc2 and WoLF PSORT into the manuscript.
I checked more pictures with the other cells. I agreed with your point and revised the part.
The correction of the errors and bad sentences was improved. However, the English language still has to be corrected (e.g. there are grammatical errors concerning the third person singular, in Abstract specifically and throughout the paper, conditional tense is used, which suggests that the authors have not proved the facts and they are only suggesting).
I will check throughout the manuscript and correct them.
line 26 - the abbreviation ABA in Abstract is not explained
Done.
Fig.5, Fig.7 - in the legend, the name of the statistic is missing
Done.
Fig.6 - in D, the statistic is missing
Done.
The English language still has to be corrected because of some elementary grammar mistakes.
Thanks for your advices. I will check throughout the manuscript and correct them.